# Reply to Standaert, B. Comment on “Postma et al. Predicted Public Health and Economic Impact of Respiratory Syncytial Virus Vaccination with Variable Duration of Protection for Adults ≥60 Years in Belgium”

**DOI:** 10.3390/vaccines11111673

**Published:** 2023-11-01

**Authors:** Maarten J. Postma, Chih-Yuan Cheng, Nasuh C. Buyukkaramikli, Luis Hernandez Pastor, Ine Vandersmissen, Thierry Van Effelterre, Peter Openshaw, Steven Simoens

**Affiliations:** 1Department of Health Sciences, Unit of Global Health, University Medical Center Groningen, University of Groningen, 9713 AV Groningen, The Netherlands; 2Department of Economics, Econometrics & Finance, Faculty of Economics & Business, University of Groningen, 9749 AE Groningen, The Netherlands; 3Janssen Pharmaceutica NV, 2340 Beerse, Belgium; 4Janssen-Cilag NV, 2340 Beerse, Belgium; 5National Heart and Lung Institute, Imperial College London, London SW3 6LY, UK; 6Department of Pharmaceutical and Pharmacological Sciences, KU Leuven, 3000 Leuven, Belgium

We have read the commentary from Baudouin Standaert [1] on our work published recently in this journal on the public health and economic impact of potential Respiratory Syncytial Virus (RSV) vaccination [2]. The objective of Postma et al. [2] was to estimate the public health and economic impact of the introduction of an RSV vaccine for older adults via a modeling study, and not to conduct a cost-effectiveness analysis (CEA) nor to estimate an acceptable vaccine price. 

RSV is one of the leading causes of acute respiratory infections (ARI) among older adults, resulting in a high disease burden [3,4,5]. With several vaccines against RSV-ARI and related diseases recently being developed [6,7,8], understanding the public health and economic impact of RSV vaccine introduction is crucial.

Against this background, we published a modeling study to predict the public health and economic impact of a potential vaccine with a multi-year duration [2], choosing Belgium as a setting considered representative of Western Europe. To ensure the external validity, we discussed and compared its results with other modeling studies assessing RSV vaccines published at the time of writing, when no RSV adult vaccines were approved nor were any assessment reports on these vaccine candidates from any national immunization authorities published.

Given that the study objective of Postma et al. [2] was to estimate the public health and economic impact, the authors took the decision-tree modeling approach to quantify the number of RSV-ARI cases prevented and, as a consequence, the number of RSV-associated hospitalizations and deaths avoided. These public health benefits were translated into economic benefits by applying healthcare utilization costs to the numbers prevented. The authors did not conduct a formal CEA. Therefore, vaccine acquisition and administration costs, typical CEA outcome using quality-adjusted life years (QALYs) lost or disability-adjusted life years (DALYs) gained, as Standaert [1] suggested in his comment, were not considered in this study. A CEA considering, all above, inputs and outcomes would be a next step in the series of analyses required to support potential vaccines’ introductions.

Standaert [1] also commented on data usage by Postma et al. [2], which we believe is merely a choice concerning the use of specific data and rendering corresponding results. Regarding the cost of hospitalization in geriatric vs. pulmonary care facilities, the data presented by Postma et al. are based on the Technical Cell for the Management of Hospital Data (TCT) [9] database of Belgium, which includes the costs of both geriatric and pulmonology wards. Other choices could be made, which might, however, be less reflective and representative of the country as a whole. As for the concern related to presenting discounted economic results, the authors also provided undiscounted results in the scenario analysis (in Supplementary Data). Presenting discounted results could be considered to support the next analytic steps toward CEAs. Regarding the concern of Standaert [1] about how deaths are accounted for in the model, we want to highlight that the projected RSV-related deaths are different between the vaccine and non-vaccine arms as a consequence of RSV cases averted by vaccination. The separate branch of overall mortality depicted in Figure 1 in the original manuscript [2] was to illustrate that in every model cycle, patients are first subject to overall mortality before accounting for RSV-related deaths with or without the vaccine’s protection. 

We agree with some of the concerns raised by Standaert [1]. In particular, with regard to age stratification, despite the efforts in one of the scenario analyses, there are still limitations in acquiring more accurate data by age subgroups (including RSV epidemiology, vaccine effectiveness, and productivity loss). Similarly, no further differentiation of indirect costs between age groups 60–64 and 65+ years could be made also due to data limitations. Additionally, how impactful RSV vaccination would be in the settings of home care and nursing homes remains unknown, as is the comparison of effectiveness with vaccines against other respiratory infections. Further comparative work is urgently required here. 

We also agree with Standaert [1] that there is a need to actively engage in partnerships between academic groups and industry, for example, in the RESCEU [10] and PROMISE [11] consortia, of which we are part. The insights from these initiatives have and will further provide valuable data to update the current modeling study, to harmonize data use in future investigations, and to design CEAs.

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
