# Peer review of "Reply to Standaert, B. Comment on “Postma et al. Predicted Public Health and Economic Impact of Respiratory Syncytial Virus Vaccination with Variable Duration of Protection for Adults ≥60 Years in Belgium”"

_vaccines, 2023, doi:10.3390/vaccines11111673_

Round 1

Reviewer 1 Report

Comments and Suggestions for Authors

***Concerning the reply from Postma and colleagues***

Alongside the general comment of why these types of analyses publish different estimates for the burden of disease, the Comment raised 6 points specific to the original analysis by Postma and colleagues:

1. The analysis does not mention vaccination costs, neither unit costs per dose nor vaccine-related costs such as administration, delivery, etc. The analysis, however, mentions cost-savings several times in the manuscript. It might well be true that considering the only the impact on disease-related costs saves money when a vaccine is deployed to reduce the burden of disease, but this is not a surprising result. What is necessary for this type of economic analysis, but also just to put the result into some context for the reader, is how much the public health authorities need to spend on a new vaccination programme in order to realise a decrease in the burden of disease. Not only did the original manuscript fail to deal with this point, the response to the Comment fails to deal with it as well. For this reason I recommend that the response goes back to the authors for further work, to either acknowledge the omission and limitation or to explain why such an omission is valid.

2. The analysis pools the population when heterogeneities are present. This is highlighted as something that the authors wished they had considered and is perhaps a limitation of the original analysis.

3. Population mortality is seemingly dealt with before individuals enter the vaccination and non-vaccination arms of the decision tree. Here, the authors don't seem to have responded directly to the issue raised in the Comment. On Figure 1 of the original analysis there appears to be three branches for the study population to travel down: (1) general population mortality; (2) non-vaccination; (3) vaccination. I believe that the Comment was hoping to get a discussion started on why the first branch appears at all, and why the general population mortality isn't a factor considered within the other two branches of the model. This should be discussed by the authors in their response.

4. No adjustment was made for the cost differences between geriatric and pulmonology hospitalisations. The response suggests that the different costs were reported as a pooled cost, which is acceptable but definitely a limitation of the original analysis, although nothing that warrants any corrections to be published.

5. Indirect costs were not separately assessed for the hospitalised 60 to 64 years age group, but the 65+ population was assumed to be retired. This is another element that the response does not deal with, so the authors should be asked to consider this before publication of their response.

6. A discount rate of 3% was used when only a cost/budget evaluation was presented by the authors. I agree that the publication of discounted and non-discounted results can aid the authors' apparent next steps in conducting a cost-effectiveness or cost-utility analysis for this vaccination programme.

Author Response

We thank the reviewer for the time and qualitative feedback on our scientific response.

  1. The analysis does not mention vaccination costs, neither unit costs per dose nor vaccine-related costs such as administration, delivery, etc. The analysis, however, mentions cost-savings several times in the manuscript. It might well be true that considering the only the impact on disease-related costs saves money when a vaccine is deployed to reduce the burden of disease, but this is not a surprising result. What is necessary for this type of economic analysis, but also just to put the result into some context for the reader, is how much the public health authorities need to spend on a new vaccination programme in order to realise a decrease in the burden of disease. Not only did the original manuscript fail to deal with this point, the response to the Comment fails to deal with it as well. For this reason I recommend that the response goes back to the authors for further work, to either acknowledge the omission and limitation or to explain why such an omission is valid.

Indeed, this research did not include vaccination costs, therefore in the author reply we have in the first paragraph clarified ‘the objective was to estimate the public health and economic impact of the introduction of a RSV vaccine for older adults via a modeling study, and not to conduct a cost-effectiveness analysis (CEA), nor to estimate an acceptable vaccine price’, and thereby the vaccination costs were not included.

To acknowledge this limitation, we clarified that vaccine acquisition and administration costs are required to support public health authority decision-making on new vaccination programmes in the future, by including in paragraph 3 of our reply that cost-effectiveness analysis can be one of the next steps. Nevertheless, the current analysis is a prerequisite for such an evaluation.

2. The analysis pools the population when heterogeneities are present. This is highlighted as something that the authors wished they had considered and is perhaps a limitation of the original analysis.

We agree with the reviewer and added in our reply that no further differentiation of indirect costs between age groups 60-64 and 65+ years could be made due to data limitation in the before last paragraph of our response.

3. Population mortality is seemingly dealt with before individuals enter the vaccination and non-vaccination arms of the decision tree. Here, the authors don't seem to have responded directly to the issue raised in the Comment. On Figure 1 of the original analysis there appears to be three branches for the study population to travel down: (1) general population mortality; (2) non-vaccination; (3) vaccination. I believe that the Comment was hoping to get a discussion started on why the first branch appears at all, and why the general population mortality isn't a factor considered within the other two branches of the model. This should be discussed by the authors in their response.

We clarified in our response that the separate branch of overall mortality depicted in Figure 1 in the original manuscript was to illustrate that in every model cycle, patients are first subject to overall mortality before accounting for RSV-related deaths with or without the vaccine’s protection.

4. No adjustment was made for the cost differences between geriatric and pulmonology hospitalisations. The response suggests that the different costs were reported as a pooled cost, which is acceptable but definitely a limitation of the original analysis, although nothing that warrants any corrections to be published.

We agree with the reviewer’s feedback, as also integrated in our initial reply.

5. Indirect costs were not separately assessed for the hospitalised 60 to 64 years age group, but the 65+ population was assumed to be retired. This is another element that the response does not deal with, so the authors should be asked to consider this before publication of their response.

We agree there are still limitations in acquiring more accurate data by age subgroups (including RSV epidemiology, vaccine effectiveness, and productivity loss) and added in our response that no further differentiation of indirect costs between age groups 60-64 and 65+ years could be made also due to data limitation.

6. A discount rate of 3% was used when only a cost/budget evaluation was presented by the authors. I agree that the publication of discounted and non-discounted results can aid the authors' apparent next steps in conducting a cost-effectiveness or cost-utility analysis for this vaccination programme.

We thank the reviewer for the feedback, and did not further amend the reply as the discounted and undiscounted results are already presented in the manuscript.

Reviewer 2 Report

Comments and Suggestions for Authors

I support publication of this response to the comment by Standaert in its current version.

Round 2

Reviewer 1 Report

Comments and Suggestions for Authors

The authors have taken the time to respond to each point raised. The manuscript is now ready for publication